# Direct Epitaxial Growth of Polar Hf_0.5_Zr_0.5_O_2_ Films on Corundum

**DOI:** 10.3390/nano12071232

**Published:** 2022-04-06

**Authors:** Eduardo Barriuso, Panagiotis Koutsogiannis, David Serrate, Javier Herrero-Martín, Ricardo Jiménez, César Magén, Miguel Algueró, Pedro A. Algarabel, José A. Pardo

**Affiliations:** 1Instituto de Nanociencia y Materiales de Aragón, Universidad de Zaragoza-CSIC, 50018 Zaragoza, Spain; barri@unizar.es (E.B.); panos@unizar.es (P.K.); serrate@unizar.es (D.S.); cmagend@unizar.es (C.M.); algarabe@unizar.es (P.A.A.); 2Laboratorio de Microscopías Avanzadas, Universidad de Zaragoza, 50018 Zaragoza, Spain; 3Departamento de Física de la Materia Condensada, Universidad de Zaragoza, 50009 Zaragoza, Spain; 4ALBA Synchrotron, 08290 Cerdanyola del Vallès, Spain; jherrero@cells.es; 5Instituto de Ciencia de Materiales de Madrid, CSIC, Cantoblanco, 28049 Madrid, Spain; riqjim@icmm.csic.es (R.J.); malguero@icmm.csic.es (M.A.); 6Departamento de Ciencia y Tecnología de Materiales y Fluidos, Universidad de Zaragoza, 50018 Zaragoza, Spain

**Keywords:** ferroelectricity, hafnium oxide, epitaxial growth, epitaxial strain

## Abstract

Single-phase epitaxial Hf_0.5_Zr_0.5_O_2_ films with non-centrosymmetric orthorhombic structure have been grown directly on electrode-free corundum (α-Al_2_O_3_) substrates by pulsed laser deposition. A combination of high-resolution X-ray diffraction and X-ray absorption spectroscopy confirms the epitaxial growth of high-quality films belonging to the *Pca*2_1_ space group, with [111] out-of-plane orientation. The surface of a 7-nm-thick sample exhibits an atomic step-terrace structure with a corrugation of the order of one atomic layer, as proved by atomic force microscopy. Scanning transmission electron microscopy reveals that it consists of grains with around 10 nm lateral size. The polar nature of this film has been corroborated by pyroelectric measurements. These results shed light on the mechanisms of the epitaxial stabilization of the ferroelectric phase of hafnia.

## 1. Introduction

Although ferroelectricity was discovered more than one century ago, it still arouses enormous scientific interest as a result of novel phenomena, materials, and applications in the fields of energy and electronics [1,2,3,4]. Among the most exciting advances of the last decade, it is worth mentioning the unexpected discovery of ferroelectricity of Si-doped hafnia (HfO_2_) films [5] in 2011. Since then, a plethora of papers have been published showing similar behaviour in many HfO_2_-based compounds [6], with the HfO_2_-ZrO_2_ solid solution being the most widely studied system [7]. Almost all the published works focus on polycrystalline films deposited between top and bottom electrodes to allow for the out-of-plane characterization of the electrical response in the resulting capacitor configuration.

The crystal structure of bulk HfO_2_ and ZrO_2_ in equilibrium at ambient conditions is that of a distorted fluorite with monoclinic space group *P*2_1_*/c*, but other symmetries (namely tetragonal *P*4_2_/*nmc* and cubic *Fm*-3*m*) are stable in different ranges of pressure and temperature, as has been well known for decades [8,9,10]. In the case of ferroelectric hafnia, there is a general consensus that it belongs to a non-centrosymmetric, metastable polymorph with orthorhombic space group *Pca*2_1_ [11], first described in bulk Mg-doped ZrO_2_ (then indexed as *Pbc*2_1_) [12]. In addition, several polar phases with orthorhombic, monoclinic, or triclinic structure have been predicted in this system [13,14,15], and a ferroelectric rhombohedral phase (space group *R*3 or *R*3*m*) has been observed in Hf_0.5_Zr_0.5_O_2_ films [16]. As a result of the subtle differences in energy among these variants, small mechanical perturbations can tip the balance in favour of one particular polymorph [9,10,14].

The heteroepitaxial deposition of thin films on single-crystalline substrates allows for the preparation of materials subject to elastic strain values of several percent units and the growth of metastable phases. These methods, called epitaxial strain engineering [17] and epitaxial stabilization [18], respectively, can give rise to the emergence of novel phases and phenomena. Epitaxial growth is thus the ideal playground for the study of hafnia-based ferroelectric films. A recent review of the literature about this topic [19] reveals that in most studies the substrate has a perovskite structure, and the bottom electrode on top of which the hafnia film is deposited is orthorhombic (La,Sr)MnO_3_ [20,21]. In a few other cases, cubic-fluorite yttria-stabilized zirconia substrates are combined with conducting oxides such as bixbyite-type Sn-doped In_2_O_3_ or Pb_2_Ir_2_O_7_ with pyrochlore structure [21,22].

In this complex crystallographic scenario, the deposition of an intermediate conducting layer has a crucial influence on the crystal structure of the growing film, thus hindering study of the intrinsic effect of the substrate on the stabilization of a particular hafnia phase. Here, we demonstrate that epitaxial Hf_0.5_Zr_0.5_O_2_ films with non-centrosymmetric structure can be grown directly on single-crystal corundum (α-Al_2_O_3_) substrates.

## 2. Experimental Section

We have focused our analysis on the Hf_0.5_Zr_0.5_O_2_ solid solution (herein denoted HZO), one of the most widely studied compositions in the hafnia-based systems, showing robust ferroelectricity in polycrystalline films, with high remanent polarization, retention, and piezoelectric response [23]. A ceramic pellet with this composition and monoclinic structure was prepared by solid-state reaction and used as a target for the pulsed laser deposition (PLD) of HZO thin films using the same experimental setup and details as described elsewhere [24]. The films were grown on corundum (α-Al_2_O_3_) single-crystal substrates from Crystec GmbH (trigonal *R*-3*c* structure with lattice parameters *a* = 4.76 Å and *c* = 12.99 Å), cut along their (0001) atomic plane (also called C-plane) with a miscut angle lower than 0.1°. The deposition was carried out at a substrate temperature of 850 °C, an O_2_ pressure of 100 mTorr, and a fluence of 1 J/cm^2^ using a KrF laser operating at a repetition rate of 10 Hz. The films were then cooled down to 20 °C at a rate of 10 °C/min in the same atmosphere. The influence of the thickness was evaluated by preparing films in a range from 1.5 to 26 nm. Their crystal structure and thickness were studied by high-resolution X-ray diffraction (XRD) and X-ray reflectivity (XRR) using a Bruker D8 Advance diffractometer equipped with parallel-beam optics and monochromatic Cu-Kα_1_ radiation (1.5406 Å wavelength). The diffractograms of all the polymorphs were simulated using the Inorganic Crystal Structure Database [25] after the structural details reported in the literature.

The local microstructure of the samples was analysed by high angle annular dark field (HAADF) imaging in scanning transmission electron microscopy (STEM) on a probe-corrected FEI Titan 60–300 microscope equipped with a high-brightness field emission gun (X-FEG) and a CEOS aberration corrector for the condenser system. This microscope was operated at 300 kV to produce a probe size below 1 Å. STEM image simulations were carried out with the Dr. Probe software package [26].

The surface morphology of the films was observed by atomic force microscopy (AFM) in a Bruker Multimode 8 apparatus using FMG01 tips from NT-MDT.

Soft X-ray absorption spectroscopy (XAS) measurements across the oxygen *K* edge were carried out in the BL29-BOREAS beamline [27] of the ALBA synchrotron (Barcelona, Spain) at room temperature, and by recording the drain current from the films placed in normal incidence configuration (i.e., with the film plane being perpendicular to the X-ray beam propagation direction) under ultra-high vacuum conditions (2 × 10^−10^ mbar). The photon flux on the samples was about 10^12^ photons/s, with an energy resolution of 50 meV. The FDMNES program [28] for multiple scattering-based XAS calculations was used to obtain the theoretical O *K* edge XAS spectra of HfO_2_, the cluster radius employed being 6.5 Å.

Pyroelectric characterization was performed to study the polar nature of the selected films, for which top Cr/Au interdigital electrodes (IDEs) with 8 μm separation between the fingers were patterned by optical lithography. A dynamic technique was used that consisted in the measurement of the thermo-stimulated current (TSC) generated in response to a low-frequency thermal wave slightly above room temperature, and its analysis to extract the pyroelectric contribution (if present). A home-made, electrically screened cell with a built-in plane-low-inductance furnace and electrical contacts for noise level below 0.01 pA was used to obtain temperature triangular waves of 8 K amplitude and ±2 K/min heating/cooling rates, while measuring sub-pA range currents with a Keithley 6514 System Electrometer (0.2 fA accuracy).

## 3. Results and Discussion

Symmetric θ/2θ X-ray diffractograms measured in all the films (Figure 1) show the sharp 0006 peak of the substrate at 2θ = 41.7° and an intense reflection of the film around 2θ ≈ 30.4°. This reflection is exclusive of non-monoclinic phases and can be indexed as 111 of cubic *Fm*-3*m*, rhombohedral *R*3 or *R*3*m*, or orthorhombic *Pca*2_1_ phase, and also as 101 of the *P*4_2_/*nmc* tetragonal space group. Finite-size, von Laue oscillations symmetrically distributed at both sides of the main peak are visible in the thinner films, revealing the crystalline coherence across the whole thickness. A shift of the main reflection towards higher angles and an increase in its width takes place upon decreasing thickness. The former observation reveals an out-of-plane shrinkage of the lattice parameters, most likely as a result of the epitaxial strain. The latter is the expected behaviour reflected in Scherrer’s equation—an inverse proportional dependency between the peak width and the size of the diffracting element. The tiny reflection at 2θ ≈ 35.5°–36° is compatible with most of the polymorphs and can be assigned to the {200} family of planes of the space groups mentioned above (including the orthorhombic, ferroelectric phase) and the monoclinic *P*2_1_*/c* phase, or to the (110) planes of the tetragonal phase. Thus, it does not provide selective information of the crystal structure, but does show that a minority fraction of the film either belongs to a different phase or to the same phase with a different orientation. For thickness above 12 nm, the main peak at 2θ ≈ 30.4° becomes less intense and less symmetric, and von Laue oscillations are missing, proving that the crystal quality worsens. A new reflection is visible in the thickest samples at 28.2° that, among the polymorphs, can only be indexed as the 111 of the *P*2_1_*/c* monoclinic structure. Thus, Figure 1 demonstrates that the HZO films between 1.5 and 12 nm on C-oriented corundum are highly epitaxial and belong to one of the non-equilibrium polymorphs. With increasing thickness, their out-of-plane lattice parameter increases (the 2θ ≈ 30.4° peak shifts to the left), and finally the bulk-like, equilibrium monoclinic phase emerges.

In the following, we will restrict our study to the 7 nm-thick film, as it shows a pure non-monoclinic structure and high crystal quality. The in-plane orientation of this sample was determined by analysing selected asymmetrical reflections. Figure 2 shows the ϕ-scan measured at 2θ = 60.4°. This reflection can be indexed only as 113 of a cubic or orthorhombic phase, thus ruling out the presence of other polymorphs in the sample. The six-fold degeneracy of the plot indicates that either the film or the substrate (or both) present twins with relative in-plane rotation of 60°. Such twins are common in single-crystal corundum substrates grown by the Czochralski technique, such as the ones used here.

The surface topography of this sample was analysed by AFM. Representative images obtained in contact mode are shown in Figure 3. Fringes about 170 nm wide and 2.1 ± 0.1 Å step high are visible, corresponding to atomic terraces of the substrate (the distance between consecutive anionic or cationic planes of corundum along its *c* axis is 2.16 Å). Inside the terraces a granular structure is found with spots of ≈10–15 nm lateral FWHM and a height of about 2 Å on average. We have performed a roughness calculation of a zoomed region of one of the terraces in Figure 3b, obtaining 0.9 Å for the arithmetic mean deviation from a perfect plane.

Complementing XRD, X-ray absorption spectroscopy (XAS) can provide additional clues to discern between the different HZO polymorphs. In transition-metal oxides, the shape, intensity, and position of the spectroscopic features that arise in the pre-edge region of O *K*-edge absorption spectra are a fingerprint of the local structural symmetry around the metal ions. In HZO the oxygen environment of the cations is different for each polymorph, producing differences in the crystal field splitting of the 3*d* levels. Specifically, in the tetragonal phase, the double degeneration of the *e_g_* orbitals is preserved, while it is lifted in the orthorhombic phase. Due to the *d*^0^ electron configuration of the Hf^4+^ and Zr^4+^ ions, the effects due to the *e_g_* orbitals in the oxygen K-edge XAS are attributable only to crystal field effects, providing a tool to distinguish between the crystallographic phases present in HZO thin films [29]. Figure 4 shows our experimental X-ray absorption spectrum of the 7 nm-thick film measured across the O *K* edge at room temperature. The two main features at approximately 533 and 537 eV, respectively, correspond to the hybridization of unoccupied O 2*p* orbitals with Hf and Zr 3*d e_g_* and *t_2g_* states. In the same figure, we also show the result of the O *K* XAS calculations we have performed for HfO_2_ under tetragonal, monoclinic, and orthorhombic crystal symmetries. The comparison with the experimental spectrum reveals a better agreement for the orthorhombic-structure case, particularly based on the peaks position, peak-to-peak, and peak-to-valley (at 534.5 eV) intensity ratios. Similar XAS studies have been reported previously for Hf_1-x_Zr_x_O_2_ films (0 ≤ x ≤ 1) [29,30,31]. All of this X-ray spectroscopic information strongly suggests that the film shows the orthorhombic, polar structure, thus providing new arguments in favour of the absence of the cubic, tetragonal, and monoclinic polymorphs.

Further structural characterization was performed by analysing plane-view and cross-sectional specimens of the 7 nm-thick HZO films by atomic resolution HAADF-STEM imaging. The plane-view observations have allowed us to extract more information about the grain size and the in-plane epitaxy relations. The low magnification image shown in Figure 5a evidences an average in-plane grain size of the order of 8–10 nm. A closer look indicates that most of the film presents crystal lattice fringes close to six-fold symmetry, as shown in Figure 5b, which corresponds to the <111> zone axis of the *Pca*2_1_ orthorhombic phase, in agreement with the XRD analysis. Interestingly, this six-fold symmetry is analogue to that of the ϕ-scan shown in Figure 2, proving a high degree of coherence between the film and the substrate, and the presence of twins rotated by 60° in the film plane. The example of Figure 5b (marked with the letter A in the figure) matches perfectly with the STEM image simulation of orthorhombic HZO along the [11-1] orientation. Another minority set of grains (such as the one marked with the letter B) presents fringes with nearly four-fold symmetry. These grains correspond to the [002] zone axis of the orthorhombic phase, matching the small reflection at 2θ ≈ 35.5°–36° visible in XRD (see Figure 1a for this thickness value). These grains are randomly oriented in the plane of the film, so they have grown epitaxially on corundum with no in-plane crystal coherence.

Figure 6a shows a representative HAADF image of the cross-sectional specimen of the 7 nm-thick HZO film. This lamella was cut along the (−1–120) plane of corundum, and the image has been collected along the [1–100] axis of the substrate. The image shows the grain boundary between two crystal twins which have grown epitaxially along the [−1–11] direction of the *Pca*2_1_ orthorhombic phase. Epitaxy relations observed in this cross-sectional view match perfectly with those observed in the planar view and with the XRD data. Crystals corresponding to the {002}-oriented grains are hard to image in the cross-sectional specimen, due to their lack of in-plane crystal coherence and their reduced grain size in comparison with the lamella width (approximately 40 nm).

The polar axis of the orthorhombic *Pca*2_1_ phase is believed to lie along the [001] direction [32], and thus 111-oriented films are expected to show a component of the spontaneous polarization in the plane of the film. We have investigated the presence of this component through pyroelectric measurements across interdigital electrodes. Varying thermo-stimulated currents between 0.035 and 0.06 pA were generated under the imposed triangular thermal wave, as shown in Figure 7. The intensity of the thermo-stimulated current, *I_TSC_*, can comprise up to three components [33]:(1)$$I_{TSC}=I_{dc}+I_{np}+I_{p}$$
where *I_dc_* is a direct current associated with the presence of the measuring device ( this always implies an input voltage, very small but not negligible when measuring signals in the sub-pA range), *I_np_* is a contribution proportional to temperature *T* that originates from different non-pyroelectric effects, such as the temperature dependence of the sample conductivity (linear in a narrow enough interval), and *I_p_* is the pyroelectric component that is proportional to the heating rate:(2)$$I_{p}=\gamma \cdot A\cdot \frac{\partial T}{\partial t}$$
where *γ* is the pyroelectric coefficient, *A* is the sample area and *t* is time. The three contributions were separated by the least square fit of the TSC to the previous equation using Matlab, and the result is shown in Figure 7. A distinctive pyroelectric component was extracted, which is also provided in the figure. Note its square wave character as expected when input is a triangular wave. A pyroelectric coefficient can be readily obtained from its amplitude.

In the film configuration with surface IDEs, this response reflects the temperature variation of an in-plane polarization component perpendicular to the electrodes, so that the relevant area is *N* × *L* × *E* where *N* is the number of fingers, *L* their length, and *E* is the film thickness. The area is then 5.7 × 10^−6^ cm^2^, from which a pyroelectric coefficient of 3.9 × 10^−5^ µC·cm^−2^·K^−1^ is obtained. This figure is significantly lower than that recently reported for HZO polycrystalline films on Si-based substrates: 5.6 × 10^−3^ µC·cm^−2^·K^−1^ for 15 nm thickness [34]. Note, however, that a spontaneous pyroelectric response is being measured in the current study, unlike in the case of [34], where the pyroelectric coefficient was measured after poling. This indicates film self-poling, which is not uncommon for films [35], and characteristic of epitaxial layers, for which strain gradients result in flexoelectric fields [36]. The coercive field of HZO films is of the order of 1 MV∙cm^−1^, so poling across 8 μm IDEs separation requires the application of high voltages which can produce the dielectric breakdown of air. Notwithstanding the possibility of increasing pyroelectric coefficients by poling, these results confirm the polar nature of the crystallographic space group.

## 4. Conclusions

Here, we demonstrate that epitaxial Hf_0.5_Zr_0.5_O_2_ films with high crystal quality and a non-centrosymmetric orthorhombic structure can be grown by pulsed laser deposition directly on C- oriented corundum substrates, as confirmed by X-ray diffraction, scanning transmission electron microscopy, X-ray absorption, and pyroelectric measurements. These samples are single-phase, contrary to most reported results, where the monoclinic polymorph is present even in epitaxial films [19]. This proves the influence of epitaxy on the stabilization of a particular, non-equilibrium phase. Our findings pave the way for further studies on the importance of epitaxial growth for the preparation of novel ferroelectric phases of hafnia, as predicted by first-principles calculations [13,14].

## Figures and Tables

**Figure 1 nanomaterials-12-01232-f001:**
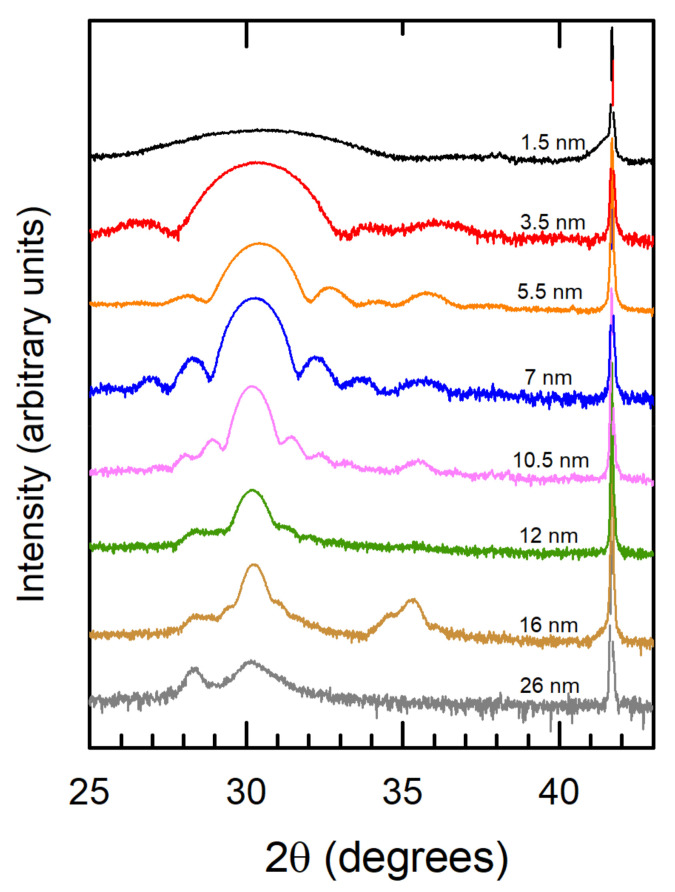
XRD of the films deposited on C-oriented Al_2_O_3_ substrates, with the thickness values indicated.

**Figure 2 nanomaterials-12-01232-f002:**
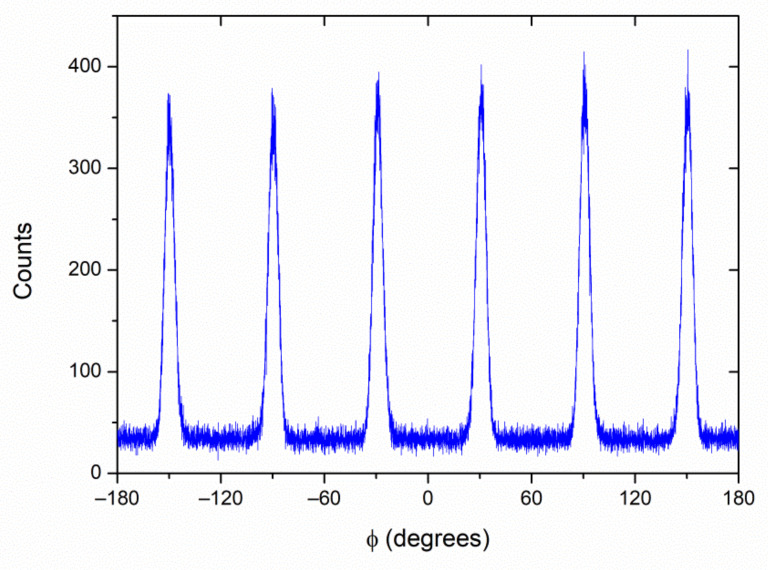
ϕ-scan measured at 2θ ≈ 60.4° in the 7 nm-thick HZO film on C-oriented corundum.

**Figure 3 nanomaterials-12-01232-f003:**
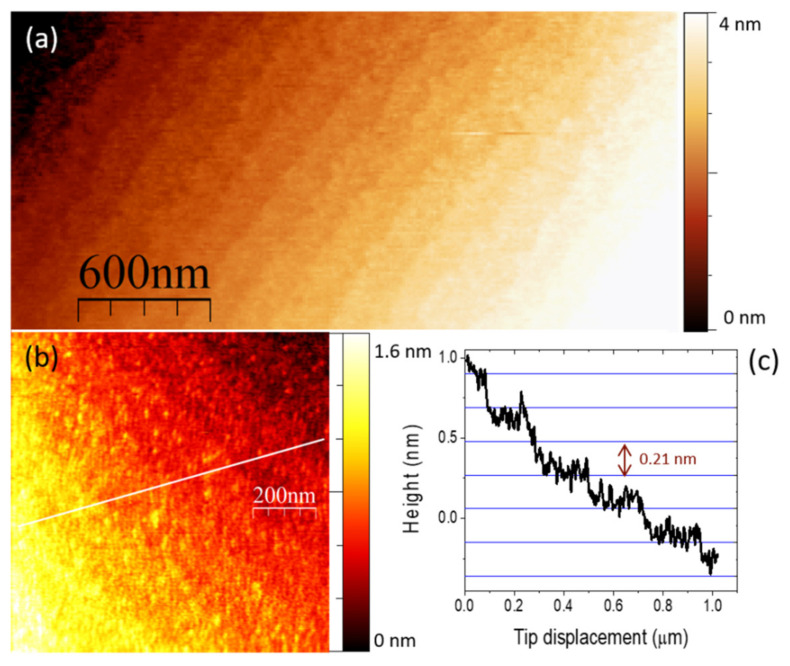
AFM (contact mode, deflection set point 25 nm) of the 7 nm-thick HZO film on C-oriented corundum. (**a**) Close up showing the stepped structure. (**b**) Detailed zoom of several terraces, and (**c**) line profile across them, showing the 0.21 nm height step. The Z-colour scale with 1.8 nm amplitude is given to the right of the image (**b**).

**Figure 4 nanomaterials-12-01232-f004:**
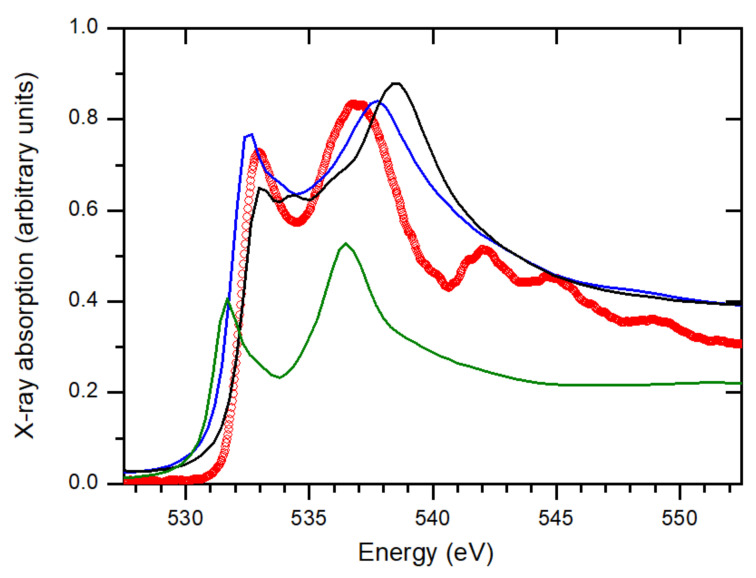
X-ray absorption spectroscopy (XAS) results. The experimental spectrum measured on the 7 nm-thick film (red circles) is compared with HfO_2_ calculations considering tetragonal (green solid line), monoclinic (black), and orthorhombic (blue) structures.

**Figure 5 nanomaterials-12-01232-f005:**
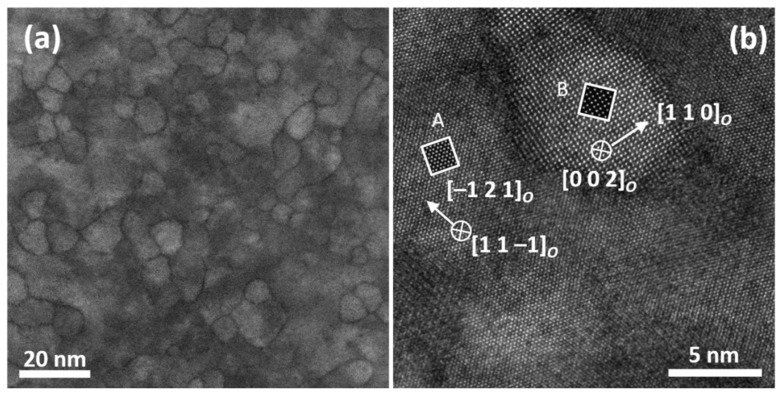
HAADF-STEM images of a planar view specimen of the 7-nm HZO thin film. (**a**) Low magnification image with a general view of the domain structure. (**b**) Detail of HZO domains indexed as orthorhombic *Pca*2_1_. See the text for explanation of the grains labelled A and B. Insets of STEM image simulations orthorhombic HZO along the corresponding orientations are superimposed on the experimental image.

**Figure 6 nanomaterials-12-01232-f006:**
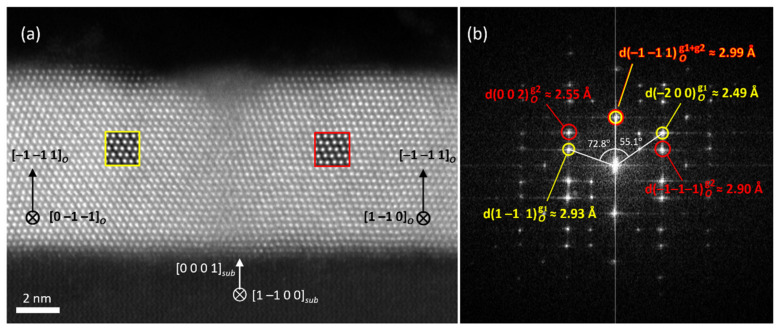
(**a**) HAADF-STEM of a cross sectional image of the 7-nm HZO thin film. Crystal orientations of the HZO domains are indexed as orthorhombic *Pca*2_1_ (marked with the subindex *O*), while the substrate orientation is indexed with the trigonal *R*-3*c* structure (subindex *sub*). Insets of STEM image simulations of orthorhombic HZO along the corresponding orientations are superimposed on the experimental image. (**b**) Fourier transform of the image shown in (**a**), indexed according to the same orthorhombic HZO crystal structure and identifying the reflections associated to the two grains shown in the image; grain 1 (to the left of the boundary, marked in yellow) and grain 2 (to the right of the boundary, marked in red). Interplanar distances and angles between reflections are those determined from the Fourier transform of the experimental image.

**Figure 7 nanomaterials-12-01232-f007:**
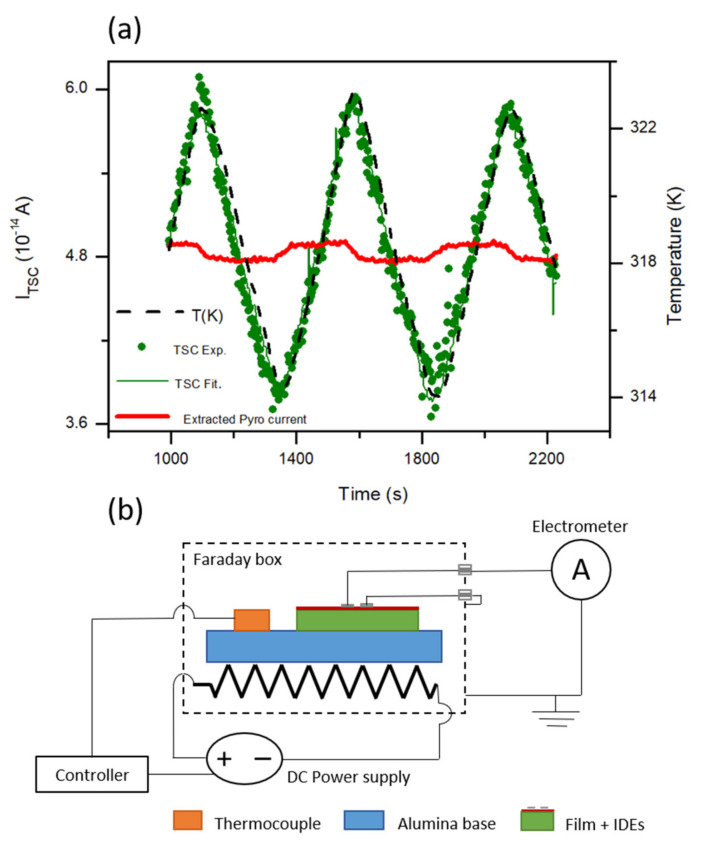
Pyroelectric measurements in the 7 nm-thick epitaxial HZO film. (**a**), Thermo-stimulated current under an imposed triangular thermal wave, and extracted pyroelectric current. (**b**) Schematic of the experimental set-up. Note that the resistor is a simplification of the heating element that consists of a low-inductance, plane furnace.

## Data Availability

Data is contained within the article.

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
