# Peer review of "Direct Epitaxial Growth of Polar Hf0.5Zr0.5O2 Films on Corundum"

_nanomaterials, 2022, doi:10.3390/nano12071232_

Round 1

Reviewer 1 Report

This article entitled Direct epitaxial growth of polar Hf0.5Zr0.5O2 films on 2
corundum is interesting and well written. Presented results seems encouraging and further research in this direction is encouraged. I recommend this paper to publish in Nanomaterials Journal. 

Reviewer 2 Report

1. From my side, regular xrd is not an effective tool for ultra-thin films, especially the thickness is as low as 1.5nm. 2. Roughness of thin films should be mentioned, the pyroelectric current is very sensitive. 3. The STEM cross-section of the thin films should be given in figure 5, and the thickness and quality of thin films can be determined directly. 4. Can you show the TEM diffraction spot pattern in figure 6? How can you determine the crystal orientations? 5. A photo picture or diagram of the circuit for pyroelectric measurement should be given in figure 7, the current is as small as dozens fa.

Reviewer 3 Report

The authors report the with high crystal quality and a non-centrosymmetric orthorhombic structure of single-phase epitaxial Hf0.5Zr0.5O2 film directly on C- oriented corundum substrates by pulsed laser deposition for applications in the fields of energy and electronics. Physical and electrical properties focus on the X-ray diffraction, X-ray absorption spectroscopy, atomic force microscopy, Scanning transmission electron microscopy, and pyroelectric measurements for the Hf0.5Zr0.5O2 films. In this study, the epitaxy on the stabilization of a particular, non-equilibrium phase, and the epitaxial growth for the preparation of novel ferroelectric phases of hafnia for predicted by first-principles calculations was innovation and reported. However, there are minor grammatical errors, and  expressions are also eccentric.

Reviewer 4 Report

In this manuscript, the authors claim that high crystal (Hf,Zr)O2 thin films with  orthorhombic phase were epitaxially grown on corumdum substrate by plused laser deposition, followed by  a series of materials characterization such as XRD, AFM, and TEM. In my opinion, the understanding of epitaxial growth by physical method, including PLD and MBE, is wrong. Firstly, there are two peaks at 35.5 and 30.4 degree in the as-grown films in 5.5-nm, 7-nm, 10.5-nm, and 16-nm HZO films. The former is corresponding to (111) of orthorhombic HZO whereas the later for (200) of monoclinic HZO or (110) of tetragonal HZO. The coexistence of two phases on the resulted films confirms that the it is not high-quality and epitaxial single crystal film. Secondly, TEM analysis in Fig 6 reveals the presence of two growth axis (<111> and <11-1>). Meanwhile, the grain boundary between them is remarkable, indicating the lower quality of HZO film. If it is high-quality single crystal film, it should be only one growth axis of [00l]. Thirdly, it is clear noted from Fig 5a that the grain size is about 8-10nm, which is conflict with that of Fig. 3, where the surface roughness is 0.2 nm. Lastly, there is a random boundary between two zones in Fig. 7, providing the proof of non-epitaxial growth in this work. To sum, this work is not novel and does not meet the standard of Nanomaterials. Therefore, my recommendation would be to reject this manuscript. Other comments are listed as followed:

  1. In Fig. 1, it is clear that the trend of the crystallinity of HZO film degrades gradually. By contrast, the peaks around 35 degree in in 16-nm sample occurs, which is similar to that of 10.5-nm one. Why?
  2. In Fig 3c, the measured height maybe not right. The step gap in distance is not distinct. It can be also attributed to the measurement error from AFM, such as intensity or phase vibration.
  3. In Fig.4, where do other two peaks ranging from 540 to 550 nm come from, besides the two main peaks below 540 nm?

Round 2

Reviewer 2 Report

Ferroelectric hafnium oxide thin film is a new and popular research material in the field of ferroelectric materials and advanced functional electrical devices. This article should be accepted. Let more researchers know the relevant research progress. An important additional reminder. The measurement and calculation of the pyroelectric coefficient is an important procedure in the measurement of the ferroelectric/optic detector. The author should ensure the accuracy of the pyroelectric current measurement.

Reviewer 4 Report

The presence of the grain boundaries between two zones (FIG. 6) must affect the electrical and thermal performances of HZO thin films. In this manuscript, the pyroelectric coefficient of of 0.39  μCm−2 K−1 is obtained, which is far lower than 48 μCm−2 K−1  for HZO in the literature (Appl. Phys. Lett. 110, 072901 (2017)). If the resulted HZO film is high-quality, and without any grain boundaries, the corresponding properties such as pyroelectric coefficient should be better. My suggestion is that this manuscript is not novel, rejected or further addtional data? 
